# The Interaction between Basic Psychological Needs, Decision-Making and Life Goals among Emerging Adults in South Africa

Eugene Lee Davids 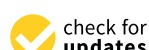

Independent Institute of Education, Varsity College, Cape Town 7700, South Africa; edavids@varsitycollege.co.za

**Abstract:** The interaction between emerging adult psychological well-being and decision-making, in South Africa, has not been explicitly explored in Self-Determination Theory. Life goals have been thought to play a role in the interaction between basic psychological needs and decision-making to promote psychological well-being. The current study, therefore, aimed to examine whether the decision-making styles employed, and the life goals which were deemed important, contribute to the understanding of the satisfaction or frustration of the basic psychological needs of emerging adults in South Africa. Data were collected cross-sectionally, using a secure, online survey among 1411 participants. The interaction between decision-making, life goals and basic psychological needs variables were examined using descriptive statistics, Pearson correlations and hierarchical regression analyses. The results in the study suggest that adaptive (vigilant) decision-making and intrinsic life goals were significant predictors for the satisfaction of the basic psychological needs. Some forms of maladaptive decision-making and extrinsic goals were predictors of the frustration of basic psychological needs. The variance explained by the various models were between 15.6–32.6%, with the results suggesting all models were significant. The results provide a novel contribution to emerging adult well-being in South Africa and Self-Determination Theory, with the implications for society, research and practice discussed.

**Keywords:** basic psychological needs; decision-making; emerging adults; goals and aspiration; self-determination theory; health; well-being

## 1. Introduction

Emerging adulthood is a distinctive developmental phase. Arnett (2007) describes it as the period between the late teenage years and the late twenties. What use to be considered a transitional phase is long enough to be considered a separate developmental period of the life course (Arnett 2011). The period of emerging adulthood (18–29 years) often is synonymous with experimenting and changes with different life experiences and decision-making in the young person's personal, relational, and occupational identity which often is stressful (Arnett 2007; Theron et al. 2021a). Emerging adulthood is more fitting than late adolescence, as the experiences of individuals in their late teenage years and twenties are uniquely different from most adolescents between ages 10 to 17 (Arnett 2007).

### 1.1. Emerging Adulthood in South Africa and the Impact on Psychological Well-Being

Young people who find themselves in the period of emerging adulthood experience an extension of instability, uncertainty, and absence of adult responsibilities; this is true for emerging adults globally, and in South Africa (Van Lill and Bakker 2022). Close to 36% of the South African population find themselves in the emerging adulthood period (Statistics South Africa 2016) where they are faced with unique challenges such as unemployment (Van Lill and Bakker 2022), poverty (De Lannoy et al. 2018), food insecurity (Gittings et al. 2021), crime (Barnes 2021), domestic violence (Gittings et al. 2021), unequal opportunities in education (Barnes 2021), substance abuse (De Lannoy et al. 2018), added to the

instability and uncertainty of this developmental period. The unique challenges faced by emerging adults exacerbate structural disadvantage, chronic poverty, and marginalization (Theron et al. 2021a), coupled with the effects of the COVID-19 pandemic, such as social, educational, and economic effects (Tomlinson et al. 2022), predispose this group of South Africans to long-term effects on their psychological well-being (Tomlinson et al. 2022). The effects on psychological well-being led to young people having to deal with feelings of fear, distress, and hopelessness which resulted in diminished psychological well-being (Tomlinson et al. 2022).

*1.2. Basic Psychological Needs Theory to Promote Psychological Well-Being*

Emerging adults in South Africa have experienced diminished psychological well-being as a result of the unique challenges they face together with the impact of COVID-19. The Basic Psychological Needs Theory (BPNT) provides a lens to promote the diminished psychological well-being experienced when young South African's basic psychological needs are satisfied. According to BPNT, a micro-theory of Self-Determination Theory, humans have psychological needs which are universal and innate, and when these needs are satisfied it promotes adjustment, development, and psychological well-being (Vansteenkiste et al. 2020). The three basic psychological needs, according to BPNT, are autonomy, relatedness, and competence. These needs have been defined as the 'psychological nutrient that is essential for individuals' adjustment, integrity, and growth' (Ryan 1995; Vansteenkiste et al. 2020). Autonomy is the ability and freedom to make informed decisions about one's actions, whereas relatedness is one's ability to feel a sense of affiliation, belonging, and connectedness and competence is one's feeling that one has successfully mastered a particular activity or task (Ryan and Deci 2020).

*1.3. Basic Psychological Needs as a Pathway for Well-Being through Decision-Making*

The three psychological needs, whether satisfied or frustrated, is integral to psychological well-being. The satisfaction of the three psychological needs promotes overall psychological well-being, whereas frustration is associated with diminished psychological well-being. In addition to the satisfaction and/or frustration of the basic psychological needs, well-being is directly impacted by the life skill of decision-making (Paez-Gallego et al. 2020). Decision-making is concerned with how individuals go about selecting an option or alternative when a choice needs to be made. The decision-making process can be informed by both rational and affective processes, which happen in isolation or synergistically (Davids et al. 2021). The interaction between emerging adults' decision-making and well-being has not been explicitly examined within Self-Determination Theory in the South African context. Numerous styles of decision-making exist, within the field of judgement and decision-making, which broadly could be categorised into adaptive and maladaptive decision-making (Davids et al. 2016b). However, Janis and Mann (1977) provide a model for decision-making in which making a decision can be categorised as being vigilant, hypervigilant, buck-passing or procrastination. Vigilant decision-making is when one examines all possible alternatives to a decisional situation and evaluates which would yield the best outcome while feeling optimistic about finding the best alternative. According to Janis and Mann's conflict model of decision-making, it is the only decision-making style which affords sound and rational decision making (Mann et al. 1997). Hypervigilant decision-making is when one makes a decision within the presence of time constraints which brings about pressure. While searching for an option, one is likely to hastily select an alternative to bring about immediate satisfaction to the decisional situation (Mann et al. 1997). The other two decision-making styles are buck-passing where one shifts the responsibility of deciding onto someone else, and procrastination where one puts off deciding until a later stage.

*1.4. Life Goals and Decision-Making as a Path toward Well-Being*

Paez-Gallego et al. (2020) have hinted to the potential role of life goals and aspirations when examining the intersection between well-being and decision-making. According to

Self-Determination Theory, when the basic psychological needs are satisfied, one would likely aspire to life goals and aspirations which are intrinsic in nature, whereas the frustration of these needs would be associated with extrinsic goals and aspirations. Intrinsic goals include personal growth, good relationships, and a sense of community, often associated with positive psychological well-being, whereas extrinsic goals are made up of wealth, fame and image, often associated with a diminished yield in psychological well-being.

This is the first South African study among emerging adults examining the role of decision-making and life goals in predicting the satisfaction or frustration of basic psychological needs, making it a novel undertaking, using the Self-Determination Theory; among South African emerging adults who experience structural disadvantage, chronic poverty and marginalization (Theron et al. 2021a) that are further exacerbated by the effects of the COVID-19 pandemic. The current study, therefore, aimed to examine whether the decision-making styles employed, and the life goals which were deemed important, contribute to the understanding of the satisfaction or frustration of the basic psychological needs as an indicator of psychological well-being for emerging adults in South Africa.

## 2. Materials and Methods

The study employed a cross-sectional design to examine the role of decision-making styles and life goals in predicting the satisfaction or frustration of basic psychological needs among emerging adults in South Africa.

### 2.1. Participants

The current study included 1411 participants with a mean age of 21.81 years (SD = 4.50; 18–33 years), who were largely female (n = 1030; 73.0%) and originated from eight of the nine provinces in South Africa, with the Gauteng (n = 802; 57%) and KwaZulu-Natal (n = 270; 19.2%) provinces being the most representative in the sample (see Table 1). The participants were all students who attended a higher education institution in South Africa, that had campuses in more than one province, to reflect the diversity of the country and the emerging adult demographic. The intended sample size was determined using the Yamane formula, where the estimated sample size was 397 emerging adults, taking into consideration both the population size of the higher education institution as well as the probability error. The final sample, in the current study, was 71.86% more than the initial sample size estimation.

**Table 1.** Demographic details.

| Variable | | n | % |
|---|---|---|---|
| **Mean Age (SD)** | | 21.81 (SD = 4.50) | |
| **Gender** | Female | 1030 | 73.0 |
| | Male | 374 | 26.5 |
| | Non-binary | 7 | 0.5 |
| **Province** | Eastern Cape | 113 | 8.0 |
| | Free State | 7 | 0.5 |
| | Gauteng | 802 | 57.0 |
| | KwaZulu-Natal | 270 | 19.2 |
| | Limpopo | 13 | 0.9 |
| | Mpumalanga | 9 | 0.6 |
| | North West | 11 | 0.8 |
| | Western Cape | 183 | 13.0 |

### 2.2. Procedure

Ethical approval was sought by the Independent Institute of Education's research ethics committee. After ethical approval was granted, the identified higher education institutions were invited to partake in the study. When permission was granted by the higher education institution, a member of staff appointed as the student manager (or an

equivalent member of staff) was contacted to inform him/her/they about the nature of the study as well as to seek permission to access the student population at the institution. Upon receiving permission to access the student population, a date and time was agreed upon between the research team and the student manager which would cause little to no disruption to the normal academic calendar and daily operations of the institution. The research team worked collaboratively with the computer applications team at the institution to make contact with the entire student population using the electronic communications database. To adhere to the protection of private information legislation in the country as well as to protect the personal information of the participants; no information or contact details of the participants were shared with the research team, but rather that the computer applications team at the institution communicated with the student population using their electronic communication channel. The participants at the higher education institution received an e-mail where information about the study was shared using both a link and QR code which directed the participants to a secure, online platform where electronic information letters, informed consent, as well as access to the secure, online study section could be accessed. To further protect the confidentiality and anonymity of the participants, all cookies and IP address collectors were disabled to ensure participant confidentiality, anonymity, as well as no tracking of the participants in the study. The secure, online study section consisted of the online consent form as well as a self-administered questionnaire, which could be completed at a time and place that would be convenient to the participant. The online questionnaire remained open for the collection of data over a two-week period, after which it was locked. The online questionnaire could be completed in approximately 20–30 min.

### 2.3. Measures

The data were collected by means of an online, self-report questionnaire where all data were safely stored in a password-protected, cloud space until the end of the study. The self-report questionnaire consisted of four sections where the following information was collected:

- *Brief demographic details.* This section of the questionnaire included brief questions related to gender, age, and province/geographical location.
- *Psychological needs.* This section of the questionnaire included 24 items from the Basic Psychological Need Satisfaction and Frustration Scale (Chen et al. 2015). The scale included 12 items that established the satisfaction of the three basic psychological needs (namely, autonomy, relatedness, and competence), and 12 items that determined the frustration of the three basic psychological needs. Participants responded to the items using a 5-point Likert scale, where 1 = not true at all and 5 = completely true. In the current study, the Cronbach alpha score for the scale was 0.857. The sub-scales generated from the scale included: autonomy satisfaction, autonomy frustration, relatedness satisfaction, relatedness frustration, competence satisfaction, and competence frustration. An example of one of the items from the scale includes: 'I feel confident that I can do things well'.
- *Decision-making.* The section assessing decision-making styles included the Melbourne Decision Making Questionnaire, which is a 22-item questionnaire based on the foundations of the Janis and Mann's conflict model of decision-making (Mann et al. 1997). The questionnaire assessed four decision-making styles, namely: vigilant, buck-passing, procrastination and hypervigilant using a 2-point Likert scale, where 2 = Not true for me and 0 = True for me. The questionnaire obtained a Cronbach alpha of 0.809, in the current study. An example of the items included in this questionnaire includes: 'When making a decision, I prefer to leave decisions to others'.
- *Life goals.* The fourth section included in the questionnaire assessed the participants life goals and aspirations using the Aspirations Index. The Aspirations Index lets participants rate the importance of each aspiration using a 4-point Likert scale, where 4 = very important and 1 = not important (Kasser and Ryan 1996). The Index gener-

ates the following sub-scales: wealth, fame, image, relationships, personal growth, community and health. These sub-scales can be further grouped into intrinsic and extrinsic goals, which will be used within this study. The Cronbach alpha for the scale in the current study was 0.781.

### 2.4. Data Analysis

All data were coded and cleaned using the Statistical Package for the Social Sciences, Version 27. After the initial coding, cleaning and screening of the data, descriptive statistics were computed to establish the means and frequencies of the variables and demographic details to be examined in the study. The descriptive statistics were followed by running inferential statistics which included computing Pearson bivariate correlations to assess associations between key study variables related to the aim of the study. The analysis continued with the use of hierarchical regression analysis where the basic psychological needs were entered as the constant/dependent variables, and the first step included the regression with the decision-making styles, and the second step in the model included the regression with both the decision-making styles and the goals. Missing data were handled using listwise deletion, a common form for dealing with missing data.

## 3. Results

### 3.1. Preliminary Analysis

The sample descriptive statistics (see Table 2) suggests that when examining the three key elements of the study, namely decision-making, goals and basic psychological needs, that for decision-making styles, the results suggest that participants were more likely to make use of buck-passing (M = 0.81; SD = 0.50) and procrastination decision-making styles (M = 0.80; SD = 0.51). In terms of the life goals and aspirations, importance was placed on both intrinsic and extrinsic goals, where intrinsic goals were deemed very important (M = 3.56; SD = 0.49) and extrinsic as being important (M = 3.28; SD = 0.34). The satisfaction of the basic psychological needs was deemed true for the participants as suggested by the results in the table, with autonomy (M = 3.75; SD = 0.80), relatedness (M = 3.95; SD = 0.88) and competence (M = 4.00; SD = 0.81) satisfaction all being categorised as true. On the contrary, the frustration of the basic psychological needs was neutral for autonomy frustration (M = 2.94; SD = 0.94) and competence frustration (M = 2.92; SD = 1.12) and somewhat true for relatedness frustration (M = 2.25; SD = 0.98).

**Table 2.** Variable descriptive statistics.

| Variable(s) | | M | SD |
|---|---|---|---|
| Decision-making styles [a] | Vigilant decision-making | 1.55 | 0.42 |
| | Buck-passing decision-making | 0.81 | 0.50 |
| | Procrastination decision-making | 0.80 | 0.51 |
| | Hypervigilant decision-making | 1.08 | 0.46 |
| Goals and aspirations [b] | Intrinsic goals | 3.56 | 0.49 |
| | Extrinsic goals | 3.28 | 0.34 |
| Basic psychological needs [c] | Autonomy satisfaction | 3.75 | 0.80 |
| | Autonomy frustration | 2.94 | 0.94 |
| | Relatedness satisfaction | 3.95 | 0.88 |
| | Relatedness frustration | 2.25 | 0.98 |
| | Competence satisfaction | 4.00 | 0.81 |
| | Competence frustration | 2.92 | 1.12 |

[a] Participants responded on a 2-point Likert scale (2 = not true for me; 0 = true for me). [b] Participants responded on a 4-point Likert scale (4 = very important; 1 = not important). [c] Participants responded on a 5-point Likert scale (5 = completely true; 1 = not at all true).

Using the cross-sectional data gathered, correlations were examined between the decision-making styles, goals and aspirations, as well as the basic psychological needs

variables (see Table 3). The results in Table 3 suggest that there are significant positive correlations between vigilant decision-making and both intrinsic and extrinsic goals. Furthermore, the results suggest that buck-passing and procrastination decision-making had significant negative correlations with both intrinsic and extrinsic goals. Hypervigilant decision-making is the only decision-making style in the study not significantly correlated with intrinsic and extrinsic goals.

**Table 3.** Zero-order correlation matrix.

| Variables | 1 | 2 | 3 | 4 | 5 | 6 | 7 | 8 | 9 | 10 | 11 | 12 |
|---|---|---|---|---|---|---|---|---|---|---|---|---|
| 1 Vigilant decision-making | 1 | | | | | | | | | | | |
| 2 Buck-passing decision-making | −0.252 ** | 1 | | | | | | | | | | |
| 3 Procrastination decision-making | −0.281 ** | 0.700 | 1 | | | | | | | | | |
| 4 Hypervigilant decision-making | −0.104 ** | 0.582 ** | 0.603 ** | 1 | | | | | | | | |
| 5 Intrinsic goals | 0.235 ** | −0.081 ** | −0.081 ** | 0.012 | 1 | | | | | | | |
| 6 Extrinsic goals | 0.071 ** | −0.026 ** | −0.016 ** | 0.001 | 0.145 ** | 1 | | | | | | |
| 7 Autonomy satisfaction | 0.306 ** | −0.329 ** | −0.362 ** | −0.298 ** | 0.231 ** | −0.021 | 1 | | | | | |
| 8 Autonomy frustration | −0.156 ** | 0.367 ** | 0.402 ** | 0.413 ** | −0.080 ** | 0.044 | −0.424 ** | 1 | | | | |
| 9 Relatedness satisfaction | 0.223 ** | −0.216 ** | −0.264 ** | −0.243 ** | 0.238 ** | −0.033 | 0.464 ** | −0.381 ** | 1 | | | |
| 10 Relatedness frustration | −0.182 ** | 0.326 ** | 0.373 ** | 0.382 ** | −0.125 ** | 0.035 | −0.372 ** | 0.481 ** | −0.640 ** | 1 | | |
| 11 Competence satisfaction | 0.323 ** | −0.447 ** | −0.454 ** | −0.360 ** | 0.186 ** | 0.035 | 0.556 ** | −0.382 ** | 0.412 ** | −0.390 ** | 1 | |
| 12 Competence frustration | −0.233 ** | 0.449 ** | 0.502 ** | 0.467 ** | −0.076 ** | 0.041 | −0.450 ** | 0.516 ** | −0.378 ** | 0.511 ** | −0.642 ** | 1 |

** Correlation is significant at the 0.01 level (2-tailed).

In addition, when examining the associations between intrinsic and extrinsic goals with the satisfaction and frustration of the basic psychological needs, the results in the table suggest that only intrinsic goals were significantly associated with the basic psychological needs. Intrinsic goals were significantly negatively associated with the frustration of the basic psychological needs, and positively associated with the satisfaction of these needs.

Examining the associations between decision-making styles and basic psychological needs, the results suggest that vigilant decision-making was significantly associated with the satisfaction of the basic psychological needs, and significantly negatively associated with the frustration of the basic psychological needs. The converse of these associations was found between buck-passing, procrastination, and hypervigilant decision-making and the basic psychological needs.

### 3.2. Main Analysis

To examine the overall aim of the study, whether the decision-making styles employed, and the life goals which were deemed important contribute to the understanding of the satisfaction or frustration of the basic psychological needs, hierarchical regression analyses were performed for each of the basic psychological needs. Tables 4–6 present the results for each of the basic psychological needs, reflecting on both the satisfaction and frustration of the basic psychological needs in each of the models.

#### 3.2.1. Autonomy Satisfaction

In Step 1, vigilant decision-making (β = 0.23; $p$ = 0.00) significantly and positively predicted autonomy satisfaction, whereas buck-passing (β = −0.08; $p$ = 0.03), procrastination (β = −0.15; $p$ = 0.00), and hypervigilant decision-making (β = −0.14; $p$ = 0.00) significantly and negatively predicted autonomy satisfaction (see Table 4, Section a). In Step 2, when adding intrinsic and extrinsic goals, all the decision-making styles remained significant predictors of autonomy satisfaction. Intrinsic goals significantly and positively predicted autonomy satisfaction (β = 0.18; $p$ = 0.00), whereas extrinsic goals was a significant, negative predictor for autonomy satisfaction (β = −0.07; $p$ = 0.00). The final model explained 22.5% of the variance for autonomy satisfaction, and the results suggest that the model was a significant predictor (F(7, 1249) = 51.769, $p \leq 0.001$) (see Table 4, Section a).

### 3.2.2. Autonomy Frustration

In Step 1, vigilant decision-making ($\beta$ = −0.08; $p$ = 0.00) was a significant, negative predictor for autonomy frustration, whereas buck-passing ($\beta$ = 0.09; $p$ = 0.00), procrastination ($\beta$ = 0.15; $p$ = 0.00), and hypervigilant decision-making ($\beta$ = 0.28; $p$ = 0.00) were significant, positive predictors of autonomy frustration (see Table 4, Section b). In Step 2, when adding intrinsic and extrinsic goals, all the decision-making styles remained significant predictors of autonomy frustration, and only extrinsic goals were a significant positive predictor of autonomy frustration ($\beta$ = 0.06; $p$ = 0.02). The final model explained 23.5% of the variance for autonomy frustration, and the results suggest that the model was a significant predictor (F(7, 1249) = 54.705, $p \leq$ 0.001) (see Table 4, Section b).

**Table 4.** Hierarchical regression analyses for Autonomy Satisfaction [a] and Frustration [b].

| Variable | Unstandardised Coefficients | | Standardised Coefficients | t | p | Variable | Unstandardised Coefficients | | Standardised Coefficients | t | p |
|---|---|---|---|---|---|---|---|---|---|---|---|
| | B | SE | β | | | | SE | B | β | | |
| Step 1 (Constant) [a] | 3.71 | 0.12 | | 31.57 | 0.00 | Step 1 (Constant) [b] | 2.03 | 0.14 | | 14.96 | 0.00 |
| Vigilant decision-making | 0.43 | 0.05 | 0.23 | 8.43 | 0.00 | Vigilant decision-making | −0.17 | 0.06 | −0.08 | −2.95 | 0.00 |
| Buck-passing decision-making | −0.13 | 0.06 | −0.08 | −2.21 | 0.03 | Buck-passing decision-making | 0.16 | 0.07 | 0.09 | 2.40 | 0.02 |
| Procrastination decision-making | −0.23 | 0.06 | −0.15 | −3.89 | 0.00 | Procrastination decision-making | 0.28 | 0.07 | 0.15 | 4.04 | 0.00 |
| Hypervigilant decision-making | −0.24 | 0.06 | −0.14 | −4.11 | 0.00 | Hypervigilant decision-making | 0.57 | 0.07 | 0.28 | 8.49 | 0.00 |
| Step 2 (Constant) [a] | 2.86 | 0.28 | | 10.33 | 0.00 | Step 2 (Constant) [b] | 1.80 | 0.33 | | 5.51 | 0.00 |
| Vigilant decision-making | 0.36 | 0.05 | 0.19 | 7.08 | 0.00 | Vigilant decision-making | −0.17 | 0.06 | −0.08 | −2.79 | 0.01 |
| Buck-passing decision-making | −0.12 | 0.06 | −0.07 | −2.01 | 0.04 | Buck-passing decision-making | 0.16 | 0.07 | 0.09 | 2.40 | 0.02 |
| Procrastination decision-making | −0.23 | 0.06 | −0.15 | −3.82 | 0.00 | Procrastination decision-making | 0.28 | 0.07 | 0.15 | 4.01 | 0.00 |
| Hypervigilant decision-making | −0.25 | 0.06 | −0.15 | −4.45 | 0.00 | Hypervigilant decision-making | 0.57 | 0.07 | 0.28 | 8.51 | 0.00 |
| Intrinsic goals | 0.41 | 0.06 | 0.18 | 6.96 | 0.00 | Intrinsic goals | −0.09 | 0.07 | −0.03 | −1.24 | 0.21 |
| Extrinsic goals | −0.17 | 0.06 | −0.07 | −2.88 | 0.00 | Extrinsic goals | 0.17 | 0.07 | 0.06 | 2.36 | 0.02 |

Note: [a] Autonomy satisfaction $\Delta R^2$ = 0.192 for Step 1; $\Delta R^2$ = 0.225 for Step 2. [b] Autonomy frustration $\Delta R^2$ = 0.231 for Step 1; $\Delta R^2$ = 0.235 for Step 2.

### 3.2.3. Relatedness Satisfaction

In Step 1, vigilant decision-making significantly and positively predicted relatedness satisfaction ($\beta$ = 0.17; $p$ = 0.00), whereas procrastination ($\beta$ = −0.09; $p$ = 0.03) and hypervigilant decision-making ($\beta$ = −0.19; $p$ = 0.00) were significant negative predictors (see Table 5, Section a). In Step 2, when adding the goals, vigilant decision-making continued to remain a significant positive predictor of relatedness satisfaction, whereas procrastination and hypervigilant decision-making were significant negative predictors. Intrinsic goals were a significant, positive predictor for relatedness satisfaction ($\beta$ = 0.19; $p$ = 0.00), and extrinsic goals were significant, negative predictors ($\beta$ = −0.08; $p$ = 0.00). The final model could be explained by a 15.6% variance, whereas the model remained significant (F(7, 1252) = 33.115, $p \leq$ 0.001) (see Table 5, Section a).

### 3.2.4. Relatedness Frustration

In Step 1, vigilant decision-making was a significant, negative predictor for relatedness frustration ($\beta$ = −0.09; $p$ = 0.00), whereas procrastination ($\beta$ = 0.14; $p$ = 0.00) and hypervigilant decision-making ($\beta$ = 0.26; $p$ = 0.00) were significant positive predictors of relatedness frustration (see Table 5, Section b). In Step 2, vigilant decision-making ($\beta$ = −0.07; $p$ = 0.01) and intrinsic goals ($\beta$ = −0.10; $p$ = 0.00) were significant, negative predictors of relatedness frustration, whereas the results suggest that procrastination ($\beta$ = 0.14; $p$ = 0.00), hypervigilant ($\beta$ = 0.27; $p$ = 0.00) and extrinsic goals ($\beta$ = 0.07; $p$ = 0.01) were all positive predictors

of relatedness frustration. Overall, the final model accounted for 20.2% variance, and the results suggest that the model was significant (F(7, 1250) = 45.219, $p \leq 0.001$) (see Table 5, Section b).

**Table 5.** Hierarchical regression analyses for Relatedness Satisfaction [a] and Frustration [b].

| Variable | Unstandardised Coefficients | | Standardised Coefficients | t | p | Variable | Unstandardised Coefficients | | Standardised Coefficients | t | p |
|---|---|---|---|---|---|---|---|---|---|---|---|
| | B | SE | β | | | | SE | B | β | | |
| Step 1 (Constant) [a] | 4.18 | 0.14 | | 30.68 | 0.00 | Step 1 (Constant) [b] | 1.47 | 0.15 | | 10.11 | 0.00 |
| Vigilant decision-making | 0.36 | 0.06 | 0.17 | 6.12 | 0.00 | Vigilant decision-making | −0.21 | 0.06 | −0.09 | −3.26 | 0.00 |
| Buck-passing decision-making | −0.02 | 0.07 | −0.01 | −0.35 | 0.73 | Buck-passing decision-making | 0.12 | 0.07 | 0.06 | 1.70 | 0.09 |
| Procrastination decision-making | −0.15 | 0.07 | −0.09 | −2.21 | 0.03 | Procrastination decision-making | 0.26 | 0.07 | 0.14 | 3.57 | 0.00 |
| Hypervigilant decision-making | −0.35 | 0.07 | −0.19 | −5.26 | 0.00 | Hypervigilant decision-making | 0.55 | 0.07 | 0.26 | 7.71 | 0.00 |
| Step 2 (Constant) [a] | 3.19 | 0.32 | | 9.90 | 0.00 | Step 2 (Constant) [b] | 1.76 | 0.35 | | 5.04 | 0.00 |
| Vigilant decision-making | 0.29 | 0.06 | 0.13 | 4.78 | 0.00 | Vigilant decision-making | −0.17 | 0.07 | −0.07 | −2.59 | 0.01 |
| Buck-passing decision-making | −0.01 | 0.07 | −0.01 | −0.11 | 0.91 | Buck-passing decision-making | 0.12 | 0.07 | 0.06 | 1.61 | 0.11 |
| Procrastination decision-making | −0.15 | 0.07 | −0.09 | −2.15 | 0.03 | Procrastination decision-making | 0.26 | 0.07 | 0.14 | 3.54 | 0.00 |
| Hypervigilant decision-making | −0.37 | 0.07 | −0.20 | −5.66 | 0.00 | Hypervigilant decision-making | 0.56 | 0.07 | 0.27 | 7.89 | 0.00 |
| Intrinsic goals | 0.48 | 0.07 | 0.19 | 7.02 | 0.00 | Intrinsic goals | −0.27 | 0.07 | −0.10 | −3.66 | 0.00 |
| Extrinsic goals | −0.20 | 0.07 | −0.08 | −2.91 | 0.00 | Extrinsic goals | 0.20 | 0.07 | 0.07 | 2.67 | 0.01 |

Note: [a] Relatedness satisfaction $\Delta R^2 = 0.120$ for Step 1; $\Delta R^2 = 0.156$ for Step 2. [b] Relatedness frustration $\Delta R^2 = 0.190$ for Step 1; $\Delta R^2 = 0.202$ for Step 2.

### 3.2.5. Competence Satisfaction

In Step 1, vigilant decision-making significantly predicted competence satisfaction (β = 0.20; $p = 0.00$), whereas buck-passing (β = −0.20; $p = 0.00$), procrastination (β = −0.18; $p = 0.00$), and hypervigilant decision-making (β = −0.12; $p = 0.00$) were all significant, negative predictors of competence satisfaction (see Table 6, Section a). In Step 2, vigilant decision-making (β = 0.17; $p = 0.00$) and intrinsic goals (β = 0.12; $p = 0.00$) were significant, positive predictors of competence satisfaction. The decision-making styles of buck-passing, procrastination and hypervigilance were all significant, negative predictors of competence satisfaction. The final model accounts for 29.3% of the variance explained, whereas the results suggest that the model is significant (F(7, 1248) = 73.747, $p \leq 0.001$) (see Table 6, Section a).

### 3.2.6. Competence Frustration

In Step 1, vigilant decision-making was a significant, negative predictor for competence frustration (β = −0.10; $p = 0.00$), whereas the other decision-making styles were significant, positive predictors (see Table 6, Section b). In Step 2, vigilant decision-making continued to be a significant, negative predictor for competence frustration (β = −0.10; $p = 0.00$), whereas buck-passing (β = 0.12; $p = 0.00$), procrastination (β = 0.22; $p = 0.00$), hypervigilance (β = 0.28; $p = 0.00$) and extrinsic goals (β = 0.06; $p = 0.01$) were significant positive predictors for competence frustration. The final model accounts for 32.6% of the variance, with the model results suggesting it is significant (F(7, 1251) = 86.401, $p \leq 0.001$) (see Table 6, Section b).

**Table 6.** Hierarchical regression analyses for Competence Satisfaction [a] and Frustration [b].

| Variable | Unstandardised Coefficients | | Standardised Coefficients | t | p | Variable | Unstandardised Coefficients | | Standardised Coefficients | t | p |
|---|---|---|---|---|---|---|---|---|---|---|---|
| | B | SE | β | | | | SE | B | β | | |
| Step 1 (Constant) [a] | 4.16 | 0.11 | | 36.83 | 0.00 | Step 1 (Constant) [b] | 1.92 | 0.15 | | 12.78 | 0.00 |
| Vigilant decision-making | 0.38 | 0.05 | 0.20 | 7.71 | 0.00 | Vigilant decision-making | −0.27 | 0.07 | −0.10 | −4.11 | 0.00 |
| Buck-passing decision-making | −0.33 | 0.06 | −0.20 | −5.79 | 0.00 | Buck-passing decision-making | 0.25 | 0.08 | 0.12 | 3.35 | 0.00 |
| Procrastination decision-making | −0.29 | 0.06 | −0.18 | −4.98 | 0.00 | Procrastination decision-making | 0.48 | 0.08 | 0.22 | 6.31 | 0.00 |
| Hypervigilant decision-making | −0.21 | 0.06 | −0.12 | −3.76 | 0.00 | Hypervigilant decision-making | 0.66 | 0.07 | 0.28 | 8.95 | 0.00 |
| Step 2 (Constant) [a] | 3.31 | 0.27 | | 12.27 | 0.00 | Step 2 (Constant) [b] | 1.66 | 0.36 | | 4.58 | 0.00 |
| Vigilant decision-making | 0.33 | 0.05 | 0.17 | 6.56 | 0.00 | Vigilant decision-making | −0.26 | 0.07 | −0.10 | −3.89 | 0.00 |
| Buck-passing decision-making | −0.32 | 0.06 | −0.20 | −5.61 | 0.00 | Buck-passing decision-making | 0.25 | 0.08 | 0.12 | 3.36 | 0.00 |
| Procrastination decision-making | −0.28 | 0.06 | −0.18 | −4.99 | 0.00 | Procrastination decision-making | 0.48 | 0.08 | 0.22 | 6.28 | 0.00 |
| Hypervigilant decision-making | −0.22 | 0.06 | −0.13 | −4.01 | 0.00 | Hypervigilant decision-making | 0.66 | 0.07 | 0.28 | 8.98 | 0.00 |
| Intrinsic goals | 0.27 | 0.06 | 0.12 | 4.78 | 0.00 | Intrinsic goals | −0.11 | 0.08 | −0.04 | −1.45 | 0.15 |
| Extrinsic goals | −0.02 | 0.06 | −0.01 | −0.40 | 0.69 | Extrinsic goals | 0.20 | 0.08 | 0.06 | 2.65 | 0.01 |

Note: [a] Competence satisfaction $\Delta R^2 = 0.280$ for Step 1; $\Delta R^2 = 0.293$ for Step 2. [b] Note: Competence frustration $\Delta R^2 = 0.321$ for Step 1; $\Delta R^2 = 0.326$ for Step 2.

## 4. Discussion

Using Self-Determination Theory, the study aimed to examine whether the decision-making styles employed, and the life goals of emerging adults in South Africa contribute to the understanding of the satisfaction or frustration of the basic psychological needs, as an indicator of psychological well-being. The results suggest that adaptive (vigilant) decision-making and intrinsic life goals and aspirations were significant predictors of the satisfaction of the basic psychological needs of autonomy, relatedness, and competence. On the contrary, the results found that less adaptive forms of decision-making (hypervigilant, buck-passing, and procrastination) and extrinsic life goals and aspirations were significant predictors of the frustration of the basic psychological needs. Overall, the findings suggest that the decision-making styles and life goals aspired toward are central to the satisfaction and/or frustration of the basic psychological needs of emerging adults in South Africa, which is synonymous with psychological well-being.

The participants were more likely to make use of maladaptive forms of decision-making, namely buck-passing and procrastination. These two forms of decision-making are often grouped as defensive-avoidant decision-making in the existing body of research (Mann et al. 1997). Although evaluating decision-making during emerging adulthood has often been deemed a complex area of research informed by numerous factors (Weinrabe and Hickie 2021), previous studies have shown that young people were more likely to engage in buck-passing and procrastination decision-making (Kornilova et al. 2018; Filipe et al. 2020; Urieta et al. 2021). Emerging adults in South Africa might have engaged in procrastination when making a decision or shifting the responsibility of making a decision as a result of the various challenges and hardships faced, such as chronic poverty, structural disadvantage and/or marginalization (Theron et al. 2021a), which often hampers psychological health and well-being. Emerging adults in South Africa experienced detrimental outcomes on their mental health and development, particularly those in more disadvantaged communities (Theron et al. 2021b). The choices we make or the alternatives selected as part of the decision-making process are often informed by the intention of the decisional situation. The intention of the decisional situation could be equated to the 'what' and 'why' of life goals and aspirations which Deci and Ryan (2000) refer to in understanding goal pursuits. The 'what' is concerned with the content of the goal or aspiration for emerging adults,

while the 'why' is the process that is engaged in as part of the goal pursuit or aspirational attainment as to whether it is controlled or autonomous in nature (Deci and Ryan 2000). The 'what' and 'why' therefore are important in understanding and predicting the outcome related to health and well-being (Deci and Ryan 2000; Sheldon and Kasser 1995). Among the sample, the life goals and aspirations saw very similar aspirations toward intrinsic and extrinsic goals. There was a slightly higher prevalence placed on the importance of intrinsic goals compared to extrinsic life goals. The differences in the importance placed on intrinsic versus extrinsic goals have been found in previous studies among emerging adults (Schmuck et al. 2000; Bespalov et al. 2017; Li and Feng 2018). From the perspective of Self-Determination Theory, persons who aspire to intrinsic life goals and aspirations are very likely to have their basic psychological needs satisfied which have been seen in the current study and shown by Vansteenkiste et al. (2006) as well as Davids et al. (2017). Emerging adults in South Africa who aspire to intrinsic life goals and engage in adaptive forms of decision-making, like the sample in the current study, are likely to have their basic psychological needs satisfied which are associated with positive psychological well-being. Resilience, among emerging adults in South Africa, becomes important to understand as a result of the unique challenges young people face in the country added with the impact of the COVID-19 pandemic, and how it could shape the adaptive forms of decision-making promoting positive psychological well-being.

The findings in the study indicate that significant associations were only found between intrinsic life goals and aspirations and basic psychological needs, whereas the opposite was found for extrinsic life goals. It is interesting to note that no significant associations were found between extrinsic life goals and basic psychological needs. Some of the potential reasons for this could be that since the study lacked an examination of the social environment which could shape the satisfaction or frustration of the basic psychological needs as being supportive, depriving or thwarting of the needs as being one reason for the current finding (Vansteenkiste and Ryan 2013). Another reason for emerging adults in South Africa could be the lack of examining of the role of resilience, which has been found to promote positive psychological well-being among disadvantaged emerging adults in South Africa (Theron et al. 2021a). It could also be questioned whether the social environment and times of uncertainty could be a possible reason for the lack of associations between extrinsic life goals and basic psychological needs frustration and/or satisfaction, particularly as the study took place during the COVID-19 pandemic. Vermote et al. (2022) have also alluded to basic psychological needs as playing a potentially less salient role during uncertain times, such as the COVID-19 pandemic, when lower order needs such as physiological and safety needs are not satisfied from a Maslowian perspective. Emerging adults and their families experienced increased situations in which their physiological (such as having a daily meal) and safety (the potential of becoming homeless) needs were put into question during the pandemic within South Africa. Interestingly, positive associations were found between intrinsic life goals and the satisfaction of basic psychological needs, which has been widely portrayed in international research on Self-Determination Theory (Davids et al. 2016a; Gunnell et al. 2014). Using the social environment and the uncertainty of the COVID-19 pandemic as a critical lens to examine the positive associations, it could be argued that 'need crafting' could be at play (De Bloom et al. 2020; Vermote et al. 2022). Need crafting is where individuals look for and engage in activities that are satisfying, which have been found in recent studies (such as Laporte et al. 2021). Negative associations were seen between intrinsic goals and the frustration of basic psychological needs. The interactions seen in the current study have been well documented in a review of the existing body of literature (such as Tang et al. 2020). The interaction between the satisfaction of the basic psychological needs saw a positive association with adaptive (vigilant) decision-making and negative associations with the frustration of the psychological needs. The converse of these associations was found for maladaptive decision-making (hypervigilant, buck-passing, and procrastination). Examining decision-making and psychological needs is a novel encounter with limited literature within Self-Determination Theory to hypothesize the interaction

seen; however, using judgment and decision-making science, vigilant or adaptive decision-making is synonymous with outcomes that promote health and well-being as well as desired outcomes (Fischhoff and Broomell 2020; Davids et al. 2015).

In the current study, the examination of emerging adults in South Africa's decision-making, life goals, and basic psychological needs using Self-Determination Theory, it could be argued that using Deci and Ryan's (2000) seminal work that examining decision-making is, in fact, the 'why' of goal pursuits (as outlined in Figure 1). The 'why' of life goal pursuits and basic psychological needs is concerned with the differentiating 'process' of goal pursuits (Deci and Ryan 2000), whereas the 'what' of goal pursuits from this perspective would be the life goal and aspirations which is focused on the 'content' of the goal and aspiration which interacts with either the satisfaction or frustration of the basic psychological needs.

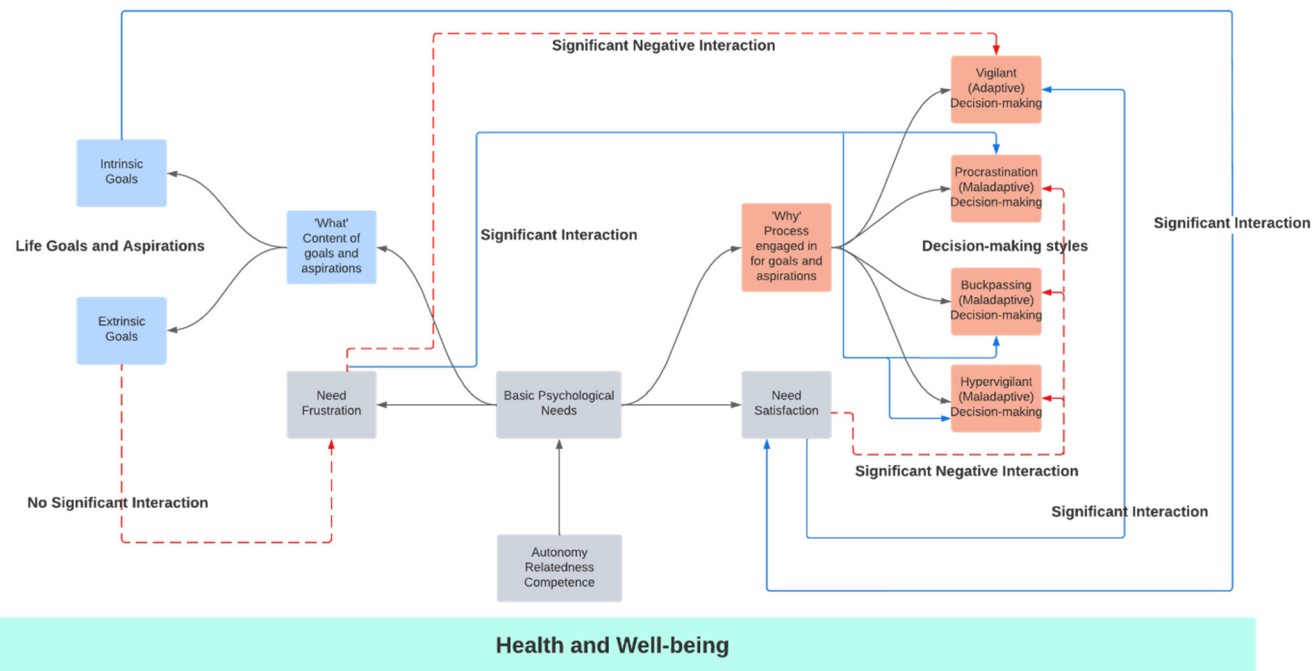

**Figure 1.** A diagrammatical representation of the interaction between basic psychological needs, decision-making and life goals of emerging adults in South Africa. All significant positive associations are indicated in the solid, blue arrow, while significant negative associations are indicated in a dash, red arrow.

Using the seminal work within Self-Determination Theory, the results in the study suggest that the satisfaction of the basic psychological needs for autonomy, relatedness, and competence were significantly predicted for the emerging adults in the study when the 'why' or rather the decision-making process was vigilant or adaptive decision-making. Vigilant or adaptive decision-making styles have previously been predictors of positive affect (Filipe et al. 2020), assertiveness (Silva et al. 2021), satisfaction with life (Filipe et al. 2020), and improved decisional self-esteem (Filipe et al. 2020) in previous studies. In the current study, it can be seen that a positive interaction was found between adaptive or vigilant decision-making and the satisfaction of the basic psychological needs which promote the health and well-being of emerging adults. Furthermore, the model suggests that the satisfaction of the basic psychological needs was also significantly predicted by the 'what' or rather the 'content' of the life goals and aspirations when they were intrinsic in nature (see Figure 1). Self-Determination Theory literature suggests that the satisfaction of the basic psychological needs interacts with the aspirations toward intrinsic goals, which is synonymous with health and well-being (Ryan and Deci 2018). These findings have

also been previously reported among young people in South Africa (Davids et al. 2016a; Roman et al. 2015).

Although decision-making is a new examination and explanation for the prediction of basic psychological needs and life goals and the aspirations of emerging adults in South Africa, what can also be learnt from the findings is that awareness, as alluded to by Ryan and Deci (2018), becomes an important theoretical component in Self-Determination Theory when examining health and well-being. Awareness is concerned with making sense of 'one's inner and outer worlds' (Ryan and Deci 2018), which has been explained as the synergistic process in the decision-making in which both the rational and affective processes in making decisions take place (Davids et al. 2021). The notion of decision-making, becomes a new variable in examining the traditional theoretical basis of the 'what' and 'why' of life goals and aspirations, as posited by Deci and Ryan (2000), and the interaction with basic psychological needs. The findings in the study help add to existing explanations for the content and process of life goals and aspirations in the pursuit of health and well-being among emerging adults as outlined in Figure 1. The examination of decision-making in Self-Determination Theory is novel, but preliminary findings from a recent study highlight the important role that decision-making can play in goals and aspirations using a brain imaging approach (Quirin et al. 2022).

### 4.1. Implications for Society, Practice and Research

The preliminary analysis suggests that the most prevalent decision-making style employed by emerging adults was procrastination and buck-passing. Interestingly, buck-passing did not significantly predict relatedness satisfaction or frustration. The explanations for this could be varied, but informed by recent developments within the field of study it could be suggested that the stress which is commonly associated with emerging adult buck-passing could be a potential explanation (Filipe et al. 2020). Young people don't have enough time or information when faced with a decisional situation. The lack of time and information could explain that the 'why' of the aspirations informing the behavioural outcome isn't always considered. The lack of the why could explain the underlying reasons for findings in the study. The inclination to make use of buck-passing decision-making among young people has also been previously explored, where it was suggested that the role of peer group and age of the young person is a significant predictor (Kornilova et al. 2018). Studying the interaction between buck-passing decision-making and well-being outcomes among emerging adults in South Africa needs to be explored further, in future studies. The results in the main analysis of the study also suggest that buck-passing decision-making among emerging adults as a non-significant variable in the predictions of relatedness satisfaction and frustration, could possibly be explained by the lack of related-ness with other persons in the social setting which could potentially hinder any attempt of buck-passing in decisional situations associated with health and well-being. Furthermore, adaptive forms of decision-making (more specifically vigilant decision-making), are seen to play an important role in health and well-being, as well as the predictions for prosocial outcomes. Interventions, both focusing on policy and programmes, should aim to promote decision-making skills which are synonymous with adaptive forms of decision-making as it is related to prosocial outcomes which promote health and well-being as seen in the current study. Interventions such as Collaborative Decision Skills Training could be considered as it has been found to promote decision-making processes, assertiveness and problem-solving skills, which predicts increased quality and satisfaction of life and well-being (Treichler et al. 2021) that could drive emerging adult health and well-being. Interventions could specifically consider the inclusion of skills-based activities (these includes role-plays, prac-tice sessions, worksheets), specific skills that drive problem-solving and planning, as well as identifying challenges that might be faced in decisional situations (Treichler et al. 2021) which hamper decision-making and life goals which would predict emerging adult health and psychological well-being.

### 4.2. Limitations and Recommendations

In examining the current study, the first limitation would be related to the generalisability of the findings. The sample included emerging adults at a higher education institution, which means the findings might not necessarily be generalizable to all emerging adults. Additionally, the sample included participants from South Africa, a developing, upper-middle-class country, with ethnic, cultural, linguistic, socio-economic and geographical diversity. Even though an attempt was made for a more heterogeneous sample, in terms of gender and geographical location, the limitation related to generalisability provides a recommendation for future studies to consider participants with various educational and diverse backgrounds. Added to the diversity reflected upon, the current study lacked collecting additional demographic details which might have described the sample better, beyond age, gender and geographic location. The second limitation relates to instrumentation and measures; only one measure was used to examine decision-making styles. A myriad of decision-making styles exists within literature (Davids et al. 2016b), and including more than one instrument to measure decision-making styles would have provided more accuracy in the interpretation of decision-making as a novel construct within Self-Determination Theory. The use of more decision-making instruments to measure decision-making styles could inform future studies to examine choice and decision-making through a Self-Determination Theory lens, adding to developments and debates within the field of study. Since, the sample was made up of more than 73% of participants who identified as female, the analyses controlled for gender. It might be worth ensuring that future studies consider a more equal or representative gender split. Additionally, all variables and constructs measured in the study were self-reported measures which is a third limitation of the study. Using more than one indicator or measurement option of variables and constructs could inform further studies and research.

### 5. Conclusions

The study attempted to examine emerging adult health and well-being using decision-making, to basic psychological needs and goal content within Self-Determination Theory. Among the emerging adults in the study, it was found that the interaction between basic psychological needs, decision-making, and life goals was explained and understood using the traditional theoretical perspective of the 'what' and 'why' of goals and aspirations. The goal content or 'what' of goals and aspirations saw an interaction between intrinsic goals with the satisfaction of basic psychological needs together with the 'why' or adaptive forms of decision-making. The study findings support the notion that basic psychological needs, decision-making, and life goals support the health and psychological well-being of emerging adults in South Africa.

**Funding:** This research received no external funding. The APC was funded by Varsity College, Independent Institute of Education.

**Institutional Review Board Statement:** The study was conducted in accordance with the Declaration of Helsinki, and approved by the Institutional Review Board of The Independent Institute of Education (IIE Reference: R.15531 on 1 July 2021).

**Informed Consent Statement:** Informed consent was obtained from all subjects involved in the study.

**Data Availability Statement:** Data presented in the paper are available on request from the corresponding author.

**Conflicts of Interest:** The author declares no conflict of interest. The funders had no role in the design of the study; in the collection, analyses, or interpretation of data; in the writing of the manuscript, or in the decision to publish the results.

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
