# Peer review of "The Interaction between Basic Psychological Needs, Decision-Making and Life Goals among Emerging Adults in South Africa"

_socsci, doi:10.3390/socsci11070316_

Round 1

Reviewer 1 Report

Review of socsci-1710202

Thank you for the opportunity to review this manuscript, Toward youth health and well-being: The interaction between basic psychological needs, decision-making and life goals.  This study uses cross-sectional data from young adults in South Africa to examine the interaction between well-being and decision-making, framed in Self-Determination Theory.

The article is clear, interesting, and easy to read, but the introduction lacks the proper foundation for why the analyses and measures were chosen. Additional details are noted below.

1.      The introduction overall provides a good review and background about BPNT, and Self-Determination Theory, but the research questions did not flow naturally from the literature presented. Instead, about half of the introduction provides background information about BPNT and SDT, but does not cite enough relevant literature that explains where the gaps are in our understanding of the role of decision-making and why these associations are important.  It would have been clearer and easier to understand if the research questions were stated explicitly, with clear evidence about how they flowed from the introduction, which seems to be a bit short.

2.       The introduction is not specific to the age range of this study, which has a mean age of 21.  There is a broad literature about the development of decision-making skills in this period of development which is not included, and could be specific to this sample (see point #5, also).

3.       Most of the methods section is clear and detailed. However, information about scale reliability and validity is notably absent, including internal reliability for the current sample.

4.       Because some the terms related to the subscales in the decision-making measure may not be widely understood by a diverse topical audience this journal may attract, further explanation of the meaning of the subscale terms would be helpful (e.g vigilant vs. hypervigilant, buck-passing, etc.).

5.       The limitation omits that the sample is 73% female. What does the literature say about how this may have impacted your regression analyses and interpretation?  

6.       Because the study took place during the pandemic, decision-making processes may have been impacted.  A review of the literature about decision-making and COVID-19 should be included either in the introduction to better frame the study, or in the discussion to provide possible explanations for the associations among variables and as a limitation. COVID-19 literature is reviewed in the discussion as it related to theories about how the pandemic may have influenced this associations but is somewhat contradictory in that the authors note the associations are the same as those found in (pre-pandemic?) literature (line 320).

7.       How was missing data handled?

8.       In the discussion, how do your findings fit in with the general literature on decision-making in young adulthood?

9.       You raise the point about individualistic and collectivistic cultures in the discussion, but this idea is not further developed or tied to other information in the manuscript. Can you expand on this and/or make the association more clear?  

Overall, the manuscript is an interesting angle on SDT and decision-making, but is based on cross-sectional data during the pandemic, which may have skewed the interpretation of results.

Author Response

Dear Editor,

Thank you for considering my manuscript for review. Below I have outlined how I have addressed the comments by the reviewers. The feedback provided has helped shape the manuscript, which I am incredibly grateful for. I hope that the revised manuscript meets the standards of the reviewers.

Reviewer 1 Comments

Response

The introduction overall provides a good review and background about BPNT, and Self-Determination Theory, but the research questions did not flow naturally from the literature presented. Instead, about half of the introduction provides background information about BPNT and SDT, but does not cite enough relevant literature that explains where the gaps are in our understanding of the role of decision-making and why these associations are important.  It would have been clearer and easier to understand if the research questions were stated explicitly, with clear evidence about how they flowed from the introduction, which seems to be a bit short.

Thank you for the valuable feedback. The introduction has been revised. I hope that the revised version satisfied the comment provided.

The introduction is not specific to the age range of this study, which has a mean age of 21.  There is a broad literature about the development of decision-making skills in this period of development which is not included, and could be specific to this sample (see point #5, also).

The feedback provided by both yourself and Reviewer 2 has been helpful. Engaging with recent literature, it was best suited to select emerging adulthood as the developmental stage, which has been incorporated and highlighted in the introduction and throughout the manuscript. Many thanks.

Most of the methods section is clear and detailed. However, information about scale reliability and validity is notably absent, including internal reliability for the current sample.

Thank you very much for the useful comment, in terms of reliability the Cronbach alpha scores for the various scales in the current study were included on page 5. The additions are highlighted in yellow.

Because some the terms related to the subscales in the decision-making measure may not be widely understood by a diverse topical audience this journal may attract, further explanation of the meaning of the subscale terms would be helpful (e.g vigilant vs. hypervigilant, buck-passing, etc.).

The introduction now provides a reflection of the key terms examined in the various sub-scales. These are highlighted on page 3.

The limitation omits that the sample is 73% female. What does the literature say about how this may have impacted your regression analyses and interpretation? 

The reflection provided on the prevalence of participants who identified as female is a very useful observation. To account for gender, the regression analyses were re-analysed to control for gender. This was a suggestion made by Reviewer 2 as well. The updated results in the analyses controlling for gender are highlighted in yellow in both the regression tables and in-text.

Because the study took place during the pandemic, decision-making processes may have been impacted.  A review of the literature about decision-making and COVID-19 should be included either in the introduction to better frame the study, or in the discussion to provide possible explanations for the associations among variables and as a limitation. COVID-19 literature is reviewed in the discussion as it related to theories about how the pandemic may have influenced this associations but is somewhat contradictory in that the authors note the associations are the same as those found in (pre-pandemic?) literature (line 320).

A reflection on the unique challenges of emerging adults in South Africa coupled with the impact of COVID-19 has been weaved into the manuscript and highlighted in yellow as suggested by Reviewer 2 as well. Thank you for the helpful suggestion.

How was missing data handled?

To outline how missing data were dealt with the following was included in the data analysis section on page 6: ‘Missing data were handled using listwise deletion, a common form for dealing with missing data.’

Complete case analysis or listwise deletion was used to handle missing data. This is where missing cases are omitted and the remaining complete cases are included in the analysis. This form of handling missing data is the most common form and was guided by the following publication: Kang H. The prevention and handling of the missing data. Korean J Anesthesiol. 2013;64(5):402-406. doi:10.4097/kjae.2013.64.5.402

In the discussion, how do your findings fit in with the general literature on decision-making in young adulthood?

A reflection on emerging adulthood to link what was discussed in the introduction has been brought into the discussion and highlighted in yellow.

You raise the point about individualistic and collectivistic cultures in the discussion, but this idea is not further developed or tied to other information in the manuscript. Can you expand on this and/or make the association more clear? 

The point on individualistic and collectivist cultures in the discussion was removed.

Overall, the manuscript is an interesting angle on SDT and decision-making, but is based on cross-sectional data during the pandemic, which may have skewed the interpretation of results.

Thank you so much for the good feedback. The role of the pandemic has been reflected upon to account for the potential role of the pandemic in the findings.

Reviewer 2 Comments

Response

This is a relevant work on decision-making, life goals and psychological needs of young people in South Africa. Yet, I suggest a set of adjustments and issues to be addressed.

The title mentions youth health and well-being but, in the text, it is not well specified what these mean, how operationalized and unclear which measures are used to measure these.

Thank you for the very helpful suggestion. The updated title now reads as: “The interaction between basic psychological needs, decision-making and life goals among emerging adults in South Africa” and some sections have been re-phrased to allow for a more clearer explanation of well-being of emerging adults.

The first sentence and the title may include South Africa as this is the gap and novelty addressed here.

Thank you for the valuable suggestion. The updated title now reads as follows: ‘The interaction between basic psychological needs, decision-making and life goals among emerging adults in South Africa’ highlighted in yellow.

The first paragraph of the paper has also been updated to reflect South Africa.

In the introduction, there is a need to convince WHY novel this topic and study, WHY South Africa and how it advances the field. I strongly suggest to add a section on SA with main characteristics, youth studies related to this one, what their context is and how this may have indirect impact on the results.

The revised version of the introduction has attempted to address the valuable contributions made in the comments to consider emerging adults and the contextual relevant of the study.

Please also list specific goals and hypotheses with analytic steps to address each.

The aim of the study was re-phrased for a more clearer examination of the specific goal of the study and the associated hypothesis.

The sample needs to be described better, what was the age range and why 21.81 as mean age may be good to classify youth? Please consult the developmental literature as above 19 these are emerging adults. Please avoid and revise “The participants, who could largely be categorised as youth” as these are youth and emerging adults. Also, why not focusing on female sample only?

The suggestion has been taken into consideration, the emerging literature speaks to emerging adults. This concept has been used throughout versus youth, which is largely contextually defined differently. In addition, the age range has also been included in the study, it might be worth highlighting that more descriptive data to describe the sample better could have been added to the results presented. This is highlighted in the limitation of the study. The study wasn’t intended to only examine the associations among females so data were not restricted only to female participants, but rather to all who voluntarily partook in the study. To consider gender, the analyses controlled for gender as recommended which have been a valuable contribution and are appreciated.  

What other sample characteristics – education, SES etc. Why mainly female above 70% and how this is controlled for?

The comment was very helpful and led to re-analysing the data by controlling for gender. The updated results are presented with in both the tables and in-text, highlighted in yellow.

What was the reliability coefficient for each scale used?

Thank you for the suggestion, the Cronbach alpha scores have been included in the methods section to reflect reliability of the scales in the current study. These additions are highlighted in the section on instruments / data collection.

I have major issues with the analytic plans and reporting of analyses. Can the authors explain the quite similar means and SD for many variables (M= .81; SD= .50 etc)?

The means and SD presented for two of the variables that are very similar are buck-passing and procrastination, these two variables form part of a category called defensive-avoidant decision-making which is a pattern of decision coping informed by the theorist who developed the scale. This could be an explanation why these mean scores are similar as they are characteristically similar in nature. A more detailed examination of this similarity can be found in the following article: Mann, L., Burnett, P., Radford, M., & Ford, S. (1997). The Melbourne decision making questionnaire: an instrument for measuring patterns for coping with decisional conflict. Journal of Behavioral Decision Making, 10(1), 1–19. https://doi.org/10.1002/(SICI)1099-0771(199703)10:1<1::AID-BDM242>3.0.CO;2-X

Table 3 and the results contain many constructs not introduced or defined in the introduction.

The introduction provides an introduction to the decision-making, basic psychological needs and life goal constructs, these additions are highlighted in the revised introduction.

Please edit carefully the text for language and style (“The variance explained by the various models were”- should be was; “The three psychological needs, whether satisfied or frustrated, is …”- should be are etc.) Please also make sure to split too long paragraphs as at times the text is hard to digest.

Thank you for the helpful comments. Additional language edits have been made.

Please explain “Using the novel examination” – how and why this is the case?

The sections which previously alluded to the novel examination have been removed from the manuscript.

I would urge to focus more on the SA context in discussing the results and providing more specific conclusions to this context and more broadly.

This has been a very useful comment in shaping the manuscript. Thank you very much. This focus has been weaved into the manuscript and highlighted in yellow in the introduction, discussion and recommendations.

Reviewer 2 Report

This is a relevant work on decision-making, life goals and psychological needs of young people in South Africa. Yet, I suggest a set of adjustments and issues to be addressed.

The title mentions youth health and well-being but, in the text, it is not well specified what these mean, how operationalized and unclear which measures are used to measure these.

The first sentence and the title may include South Africa as this is the gap and novelty addressed here.

In the introduction, there is a need to convince WHY novel this topic and study, WHY South Africa and how it advances the field. I strongly suggest to add a section on SA with main characteristics, youth studies related to this one, what their context is and how this may have indirect impact on the results.

Please also list specific goals and hypotheses with analytic steps to address each.

The sample needs to be described better, what was the age range and why 21.81 as mean age may be good to classify youth? Please consult the developmental literature as above 19 these are emerging adults. Please avoid and revise “The participants, who could largely be categorised as youth” as these are youth and emerging adults. Also, why not focusing on female sample only?

What other sample characteristics – education, SES etc. Why mainly female above 70% and how this is controlled for?

What was the reliability coefficient for each scale used?

I have major issues with the analytic plans and reporting of analyses. Can the authors explain the quite similar means and SD for many variables (M= .81; SD= .50 etc)?

Table 3 and the results contain many constructs not introduced or defined in the introduction.

Please edit carefully the text for language and style (“The variance explained by the various models were”- should be was; “The three psychological needs, whether satisfied or frustrated, is …”- should be are etc.) Please also make sure to split too long paragraphs as at times the text is hard to digest.

Please explain “Using the novel examination” – how and why this is the case?

I would urge to focus more on the SA context in discussing the results and providing more specific conclusions to this context and more broadly.

Author Response

(The authors gave the same response as above.)

Round 2

Reviewer 2 Report

Thanks for the detailed explanations and revisions. Please be careful in language editing and typos.